# Genome-wide analyses across Viridiplantae reveal the origin and diversification of small RNA pathway-related genes

Sibo Wang[1,2,6], Hongping Liang [1,3,6], Yan Xu[1,3], Linzhou Li [1,4], Hongli Wang[1,3], Durgesh Nandini Sahu[1], Morten Petersen[2], Michael Melkonian[5], Sunil Kumar Sahu [1✉] & Huan Liu [1,2✉]

Small RNAs play a major role in the post-transcriptional regulation of gene expression in eukaryotes. Despite the evolutionary importance of streptophyte algae, knowledge on small RNAs in this group of green algae is almost non-existent. We used genome and transcriptome data of 34 algal and plant species, and performed genome-wide analyses of small RNA (miRNA & siRNA) biosynthetic and degradation pathways. The results suggest that Viridiplantae started to evolve plant-like miRNA biogenesis and degradation after the divergence of the Mesostigmatophyceae in the streptophyte algae. We identified two major evolutionary transitions in small RNA metabolism in streptophyte algae; during the first transition, the origin of DCL-New, DCL1, AGO1/5/10 and AGO4/6/9 in the last common ancestor of Klebsormidiophyceae and all other streptophytes could be linked to abiotic stress responses and evolution of multicellularity in streptophytes. During the second transition, the evolution of DCL 2,3,4, and AGO 2,3,7 as well as DRB1 in the last common ancestor of Zygnematophyceae and embryophytes, suggests their possible contribution to pathogen defense and antibacterial immunity. Overall, the origin and diversification of DICER and AGO along with several other small RNA pathway-related genes among streptophyte algae suggested progressive adaptations of streptophyte algae during evolution to a subaerial environment.

[1] State Key Laboratory of Agricultural Genomics, BGI-Shenzhen, Shenzhen, China. [2] Department of Biology, University of Copenhagen, Copenhagen, Denmark. [3] BGI Education Center, University of Chinese Academy of Sciences, Shenzhen, China. [4] Department of Biotechnology and Biomedicine, Technical University of Denmark, Lyngby, Denmark. [5] Integrative Bioinformatics, Department Plant Microbe Interactions, Max Planck Institute for Plant Breeding Research, Cologne, Germany. [6] These authors contributed equally: Sibo Wang, Hongping Liang. ✉email: sunilkumarsahu@genomics.cn; liuhuan@genomics.cn

Small RNAs (sRNAs) are distinct genetic and epigenetic regulators in various organisms, involved in the modification of DNA and histone methylations to the modulation of the abundance of coding or non-coding RNAs[1,2]. In plants, sRNAs are categorized into two major classes, microRNA (miRNA) and small interfering RNA (siRNA)[3–9]. siRNAs are the major regulatory sRNAs, which primarily engage in RNA silencing (Fig. S1). Many of the characterized sRNAs are involved in the regulation of diverse biological programs, processes, and pathways in response to developmental cues, environmental signals/stresses, or pathogen infection[4,10].

miRNAs are the functionally most important and studied class of sRNAs in plants, and their metabolism, especially their biogenesis, is a complicated process (Fig. S1). In plants, miRNAs are processed from short hairpin precursors encoded by *MIR* genes. Firstly, the *MIR* gene is transcribed by RNA polymerase II (Pol II) to produce the pri-miRNAs (primary transcripts of *MIR*s). Then, the pri-miRNAs are recognized and processed by Dicer-like RNase III endonucleases (DCLs) into a miRNA/miRNA* duplex with a length of ~21 nucleotides (nt). In *Arabidopsis thaliana*, there are four DCLs (DCL1-4) regulating miRNA production[11,12]. Especially, DCL1 catalyzes the production of most miRNAs with the assistance of accessory proteins including the double-stranded RNA-binding protein Hyponastic Leaves 1 (HYL1) and the zinc-finger protein Serrate (SE) in the miRNAs biogenesis process[13,14]. Subsequently, the nascent miRNA/miRNA* duplex generated by DCLs is methylated by the small RNA methyltransferase HUA Enhancer 1 (HEN1) (Fig. S1)[15]. Methylated miRNA/miRNA* duplexes are further processed by ARGONAUTEs to form the RNA-induced silencing complex (RISC) in the nucleus, where one strand of the duplex associates with ARGONAUTE1 (AGO1) and another strand is ejected and degraded[16,17]. *Arabidopsis* has 10 AGO proteins, and previous studies have shown that miRNAs guide AGO1

to the target RNA[18]. The assembled RISC is then exported to the cytosol by EXPO proteins and is guided to repress target mRNAs by the recognition of miRNAs[15,19]. HEN1 SUPPRESSOR1 (HESO1) and RNA URIDYLYLTRANSFERASE1 (URT1) are nucleotidyl transferases that act cooperatively to 3′ oligouridylate unmethylated miRNAs, thereby triggering miRNA decay. SMALL RNA DEGRADING NUCLEASE (SDN) is a family of 3′–5′ exonucleases that also participate in miRNA degradation (Fig. S1).

Endogenous siRNAs are also able to function in the silencing of transposable elements and repeat sequences to maintain genome stability through an RNA-directed DNA methylation mechanism (RdDM)[20]. Single-stranded RNAs (ssRNA) are generated from RdDM target loci by the activity of RNA polymerase IV (Pol IV). The ssRNAs are then converted to double-stranded RNAs by RNA-DEPENDENT RNA POLYMERASE 2 (RDR2), and the double-stranded RNAs are processed by DCL3 to 24-nt siRNA duplexes, which undergo HEN1-mediated methylation (Fig. S1). At last, the mature siRNA incorporates into AGOs, which instigates an interaction at the target loci with the transcribed scaffold RNAs mediated by RNA polymerase V (Pol V). This combination results in the silencing of the target loci as a part of a DNA methylation process[1,6,9].

Based on fossil evidence and molecular estimates, the land plant (embryophyte) flora originated 450–500 million years ago (Mya)[21–23]. Transition to land represents one of the most important steps in the evolution of life on Earth, ultimately leading to the formation of current terrestrial ecosystems. However, crucial steps in green plant terrestrialization have not been fully understood. Viridiplantae comprise two major clades: Streptophyta, which contains streptophyte algae and land plants (embryophytes), and Chlorophyta, which includes all remaining green algae (with exception of Prasinodermophyta, which split before the divergence of Streptophyta and Chlorophyta)[24] (Fig. 1). The streptophyte algae

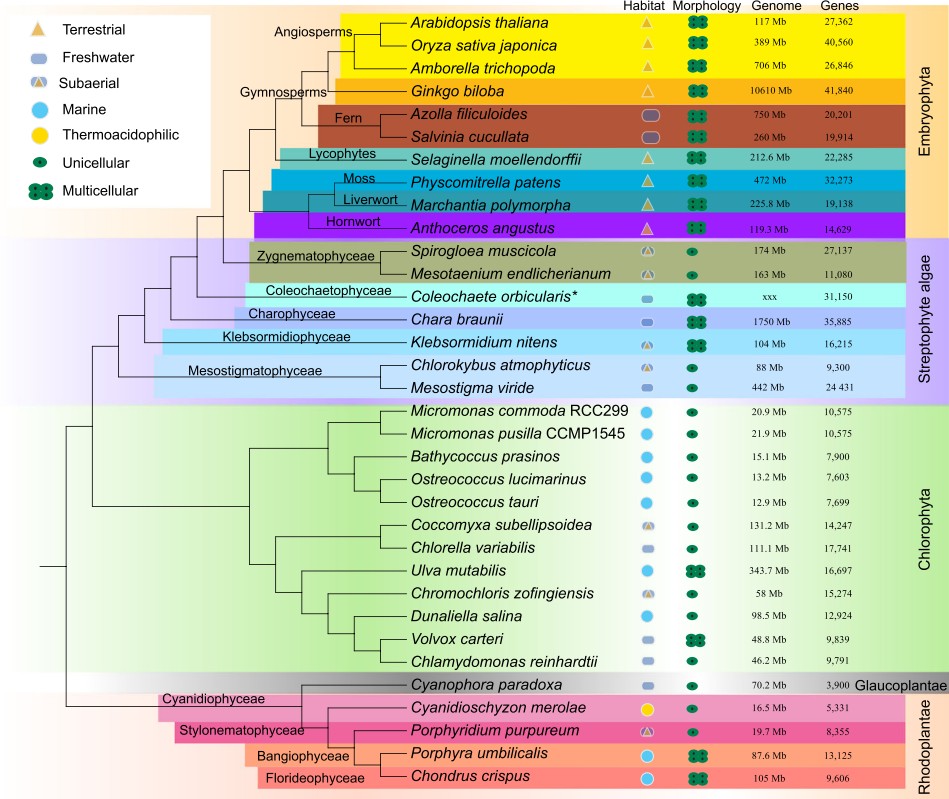

**Fig. 1 Phylogeny of Archaeplastida showing the species included in this study.** For each species, the genome size and total gene number as well as habitat and morphological characteristics are also indicated. The tree was modified and compiled from refs. [10,23,40]. * denotes transcriptome data.

comprise a grade of lineages with class Zygnematophyceae being sister to embryophytes[21,23,25–28]. Environmental stress tolerance of streptophyte algae has been implicated to have facilitated their transition to land[21,29–32]. During the whole terrestrialization process, streptophyte algae faced various environmental stresses; furthermore, they obtained novel genes by horizontal gene transfer from bacteria as well as through gene duplications, gene family expansions, and diversification. Since many genes are involved in the synthesis and regulation of sRNAs, they likely played a key role in the evolution of responses to developmental cues, and environmental stresses during plant terrestrialization[1,19,33].

In recent years, with the development of high throughput sequencing technology, the miRNA database from green algae to angiosperms has expanded significantly. This information has led to the recognition of sets of conserved miRNA families in embryophytes[19,33]. But it has also highlighted the lack of conservation of miRNAs between green algae and embryophytes indicating multiple independent origins of *MIR* genes, and complex evolutionary histories of the respective genes. The increasing availability of streptophyte and chlorophyte algal genomes, provides ample opportunity to shed light on the evolution of the sRNA metabolic pathways[21,29,34].

In this study, we used genome-wide analyses to investigate the molecular evolution of small RNA pathways across Archaeplastida, specifically focusing on genes involved in the biogenesis and degradation processes of sRNAs during plant terrestrialization (i.e., from streptophyte algae to embryophytes). Our genome-wide identifications coupled with phylogenetic analyses allowed us to gain insight into the step-wise origin and evolutionary diversification of genes involved in the small RNA machinery.

## Results

### Phylogenetic analysis of the DCL and AGO genes across Viridiplantae.
DCL and AGO are the two key factors regulating sRNA metabolism, therefore, we firstly identified the homologs of *DCL* and *AGO* from the published Archaeplastida genomes (Supplementary data S1, S2) and the phylogenetic analyses were performed with the retrieved DCL and AGO genes from these species. Previous phylogenetic analyses of DCL genes across embryophytes displayed four DCL clades relying on the *Arabidopsis* DCL1-4[12,34] but our phylogenetic trees using maximum likelihood and Bayesian methods suggested the existence of two additional clades (DCL-New and DCL-Algae; Figs. 2, S2, S3). The earliest-diverging streptophyte lineage, the Mesostigmatophyceae (*M. viride* and *C. atmophyticus*), encoded only the algal-type DCL (DCL-Algae; Fig. 2a). The phylogenetic tree also indicated that DCL1 and other DCLs presumably evolved after the divergence of the Mesostigmatophyceae in the last common ancestor of Klebsormidiophyceae and all other Streptophyta (Fig. 2a). Surprisingly, we could not identify DCL1 in the two published genomes of Zygnematophyceae, which could be due to a gene loss event in the ancestor of the Zygnematophyceae. In order to further test the phylogeny, we performed an additional phylogenetic analysis by including sequences from the 1kp dataset[35]. The phylogenetic tree showed, that all DCL proteins from Zygnematophyceae were distributed in either DCL2/3/4 or DCL-New, thus corroborating our conclusion (Fig. S4). DCL1 seems to be separated from other DCL clades although bootstrap support values for all DCL clades were generally low (Fig. 2a), which is in accordance with previous analyses[34,36]. Our phylogenetic analysis also suggested that DCL2/3/4 expanded and diverged in the embryophytes[34,36]. The genome-wide analysis revealed that of the streptophyte algae only the genomes of Zygnematophyceae possess ancestral forms of DCL2/3/4 genes (Fig. 2a). *Mesotaenium endlicherianum* contained 1 copy, while 3 copies of the ancestral form of DCL2/3/4 were

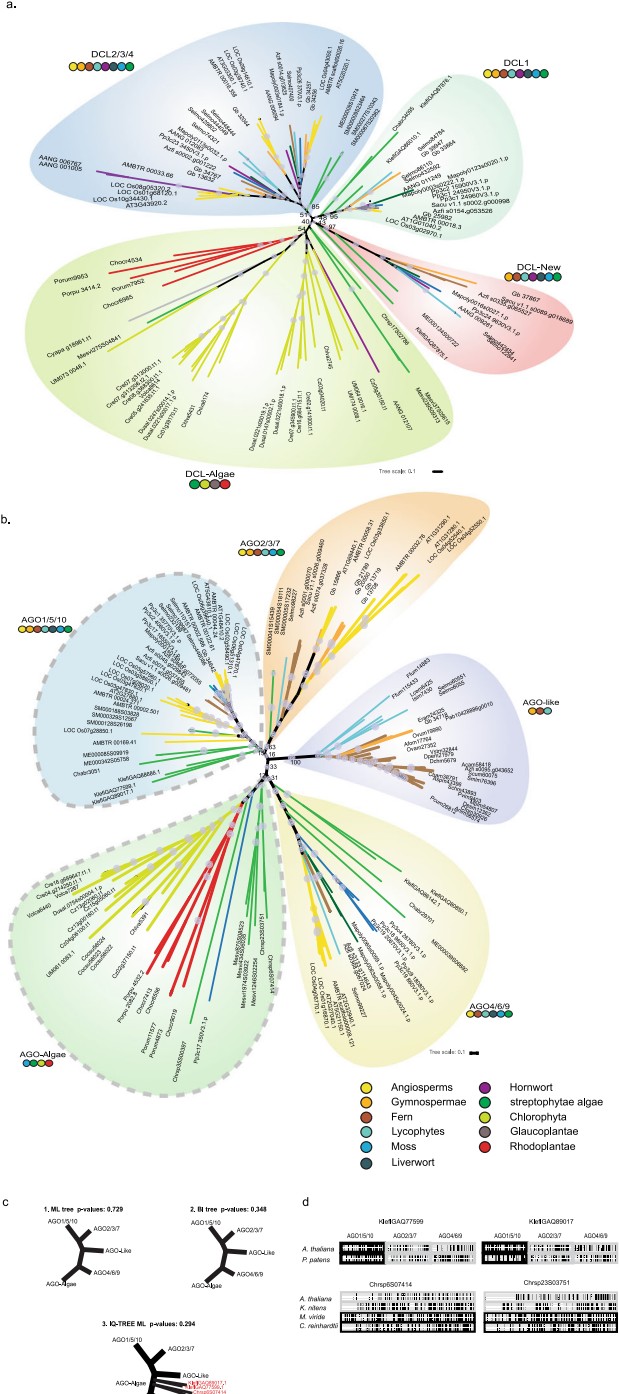

**Fig. 2 Phylogenetic trees of AGO and DCL involved in small RNA pathways across the Archaeplastida. a** A maximum likelihood tree for DCL genes in selected Archaeplastida species. **b** A maximum likelihood tree for AGO genes in selected Archaeplastida species. Bootstrap values (500 replicates) are shown on some internal branches. The ID to corresponding species names is listed in Supplementary data S3. **c** AU test to check the most reliable phylogeny among RAxML, Bayesian and IQ trees (for details see "Methods" and Supplementary data S5). **d** Sequence alignment of the ancestral AGO from *K. nitens* and *C. atmophyticus* with AGO from representative species.

detected in the *Spirogloea muscicola* genome, in accordance with its genome triplication[21]. In addition, our phylogenetic tree displayed a novel DCL clade (DCL-New; Fig. 2a). Intriguingly, the novel DCL clade comprised only genes from streptophyte algae

a.

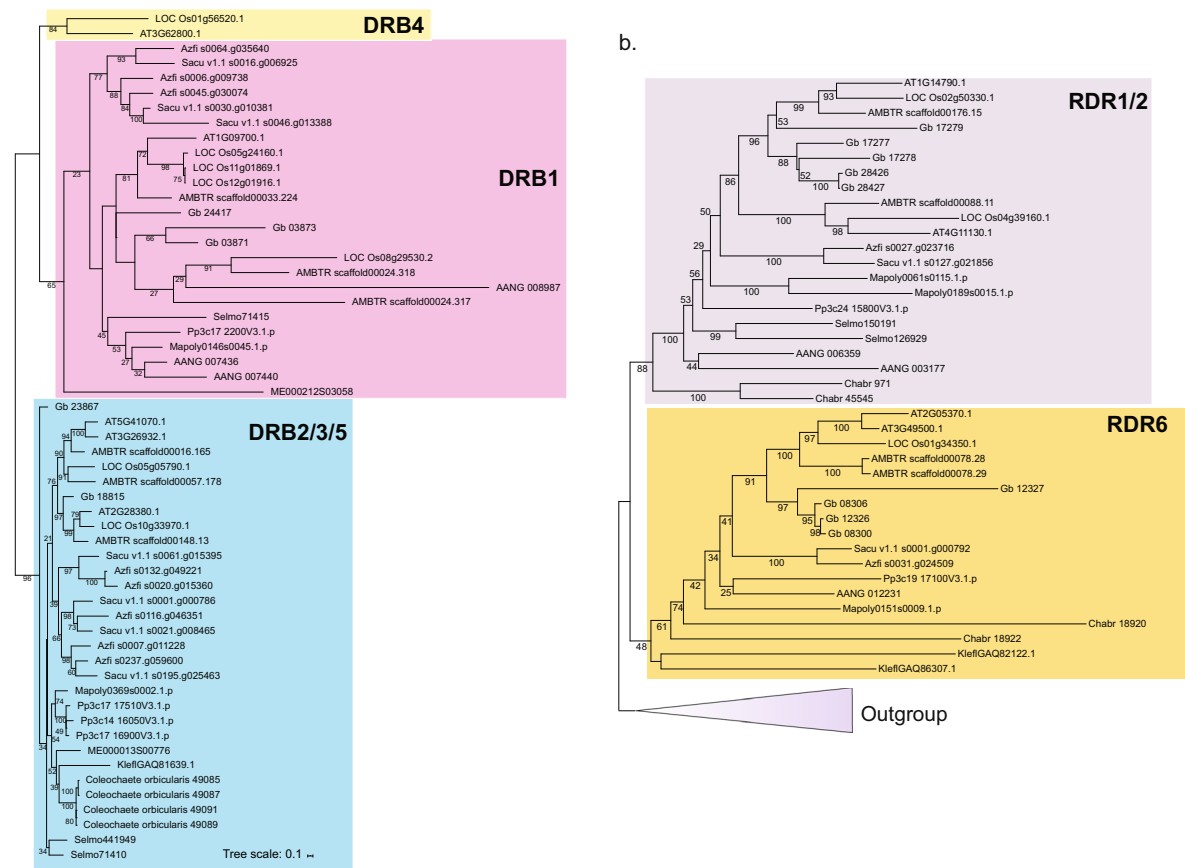

**Fig. 3 Phylogenetic analyses of the DRB and RDR genes. a** A maximum likelihood tree of DRB in selected Viridiplantae species. **b** A maximum likelihood tree of RDR in selected Viridiplantae species. Other RDR homologs were taken as an outgroup. The tree was constructed by maximum likelihood using 500 bootstrap replicates, values ≥20% are shown on internal branches. For species designations of sequence IDs see Supplementary data S3.

(*K. nitens* and *M. endlicherianum*), bryophytes, lycophytes, ferns, and gymnosperms but no angiosperms (Fig. 2a).

AGO genes in embryophytes can be classified into four main clades based on our maximum likelihood and Bayesian trees, AGO 1/5/10, AGO 2/3/7, AGO 4/6/8/9, and AGO-like[18] (Fig. 2b, Figs. S5, S6). Most of the angiosperms only contain the former three clades, and different AGO proteins exhibit distinct functions by binding to different groups of small RNAs[34]. Our phylogenetic tree largely corroborated previous analyses[36–38] (Fig. 2b, Figs. S5, S6). More importantly, we gained novel insight into the evolutionary origin of each clade of AGO genes: the early-diverging streptophyte algal class Mesostigmatophyceae only contained algal-type AGO genes (Figs. 2b, S5, S6). The AGO 4/6/8/9 clades (represented as a single group hereafter) were found to be more closely related to the algal-type AGOs than the other AGO clades. Two copies of AGO genes from the *K. nitens* genome as well as AGO genes from *Chara braunii* and *Mesotaenium endlicherianum* were identified in this group, indicating that the ancestral form of the AGO 4/6/8/9 genes originated after the divergence of the Mesostigmatophyceae but before divergence of Klebsormidiophyceae (Fig. 2b, Figs. S5, S6). The same was true for the AGO 1/5/10 clade, in which three AGO genes from *K. nitens*, one gene from *C. braunii*, two genes from *M. endlicherianum*, and three genes of *Spirogloea muscicola* were found. Interestingly, for both AGO clades, AGOs from Zygnematophyceae constituted the closest relatives to embryophytes in the phylogenetic tree, which is in accordance with the newly established sister clade relationship

between Zygnematophyceae and embryophytes. Despite testing different methods, including Maximum Likelihood and Bayes, using the best predicted model of evolution and different software (RAxML, MrBayes and IQtree), support values at key nodes in the different phylogenetic trees were relatively low (Fig. 2b, c, Figs. S5, S6), perhaps indicating intrinsic sequence variation in the ancestral plant-like AGO with respect to Chlorophyta, and the mature AGO (of embryophytes). The clade AGO 2/3/7 was only found in *Spirogloea muscicola* (Fig. 2b). Coleochaetophyceae, Charophyceae, and Klebsormidiophyceae do not seem to encode this group of AGO genes (Supplementary data S3, Fig. 2b, Figs. S5, S6), suggesting that clade AGO 2/3/7 evolved in the last common ancestor of Zygnematophyceae and embryophytes. Because of the ambiguous position of four ancestral AGO sequences from the early branching Streptophyta (*Chlorokybus*, *Klebsormidium*) in the IQ-tree (Fig. S5), we performed an approximately unbiased (AU) test (see "Methods", Fig. 3c), and sequence identity comparisons to test the topology and reliability of the phylogenetic trees (Fig. 3d), and found that the phylogeny by RAxML and MrBayes were more reliable than that of the IQ tree (see the *p*-values in Fig. 2c). In addition, the identity of two ancestral plant-like AGO sequences (KleflGAQ77599 and KleflGAQ89017) was closest to plant AGO 1/5/10 (Fig. 2d). Moreover, the two *Chlorokybus* sequences (Chrsp23 S03751 and Chrsp6S07414) were closer to the AGO-algae sequences (i.e., *Mesostigma viride* and *Chlamydomonas reinhardtii*) than to other streptophyte algal sequences (i.e., KleflGAQ77599) (Fig. 2d). Genome-wide analyses also revealed

that the long-branch AGO-like clade only occurred in lycophytes, ferns, and gymnosperms (Fig. 2b; Figs. S5, S6), its absence in bryophytes and angiosperms currently remains elusive.

**Genome-wide analysis of the distribution of other miRNA pathway-related genes**. A previous study indicated that *HEN1*, *HESO1*, *URT1*, and *SDN* existed in the most recent common ancestor of embryophytes as single-copy genes, raising the possibility that they are orthologues in Viridiplantae (Supplementary data S3)[34,36]. In order to comprehensively understand the evolutionary origin of the sRNA metabolism (biogenesis and degradation), here we made further genome-wide investigations on the distribution of each gene across Archaeplastida (Supplementary data S2, S3). *Serrate* (SE) was identified in all taxa of Viridiplantae, even Rhodoplantae encoded the SE protein. Furthermore, SE existed as a single copy gene in algae but underwent duplication in some embryophytes (mosses, ferns), while other embryophytes (*Gingko*, rice) displayed multiple copies (Fig. S7). *HEN1* was only detected in Viridiplantae and seemed to have been lost in some lineages of Chlorophyta (Mamiellophyceae, Trebouxiophyceae, unicellular, flagellate Chlorophyceae such as *C. reinhardii* and *D. salina* [Fig. S8]). Except for *A. thaliana* and *S. muscicola*, it occurred exclusively as a single copy gene (Fig. S8). Similar to *HEN1*, *HESO1* was identified only in Viridiplantae, and it could be detected in multiple copies in some Chlorophyceae as well as in *Oryza* (Fig. S9). URT1 was found in a single copy form in all the taxa of Archaeplastida. It was apparently lost in the Mamiellophyceae (Fig. S10 and Supplementary data S3). The ancestral SDN might have evolved in the common ancestor of the Viridiplantae as the two major lineages of SDNs (SDN 5 and SDNs 1–3) are present in both Chlorophyta and Streptophyta including Mamiellophyceae but are absent in Glaucoplantae and Rhodoplantae (Supplementary data S3) (Fig. S11).

During the biogenesis of microRNAs (miRNAs) in the nucleus, the double-stranded RNA-binding (DRB) protein could assist DCL1 in the precise processing of primary miRNAs (pri-miRNAs) to miRNAs[39]. There are three groups of DRBs, namely DRB2/3/5, DRB1, and DRB4[39], and our results were in agreement with this classification (Fig. 3a). But, for the first time, we identified the presence of DRB2/3/5 homologs in *K. nitens* and later-diverging streptophyte classes (Coleochaetophyceae, Zygnematophyceae) except *C. braunii* (Fig. 3a). A DRB1 homolog was found in Zygnematophyceae (*M. endlicherianum*) but not in other streptophyte algae (Fig. 3a), suggesting that it might have originated in the last common ancestor of Zygnematophyceae and embryophytes. The third group corresponding to DRB4 was detected in derived angiosperms, implying its recent diversification (Fig. 3a, Supplementary data S3).

In the chlorophyte green alga (*C. reinhardtii*), miRNA-induced gene silencing is very similar to that of higher land plants, particularly in terms of the activation of the RNA-induced silencing complex and mRNA targeting[40]. In contrast to animals, plant miRNAs require near-perfect complementarity to direct the cleavage of target mRNAs. In plants, *MIR* genes are embedded within the noncoding sequences between protein-coding genes, while in Chlamydomonas, *MIR* genes are embedded within the introns of protein-coding genes and transcribed by Pol II, similar to animal MIR (plesiomorphy)[41]. This difference in the miRNA machinery between green algae and embryophytes could be due to the difference in habitat, various other environmental adaptations, or structural and developmental complexity. Moreover, our genome-wide identification of all genes involved in miRNA metabolism, collectively suggests that Viridiplantae started to form a complete mechanism of plant-like miRNA

biogenesis and degradation after the divergence of the Mesostigmatophyceae in the streptophyte algae. Although many Chlorophyta, including *C. reinhardtii* were reported to have miRNA[42,43], the absence of genes involved in miRNA biogenesis and degradation processes homologous to Streptophyta, suggests a different miRNA metabolism in Chlorophyta.

**The evolution of siRNA biogenesis-related RNA polymerases across Viridiplantae**. siRNA-mediated chromatin modification plays a pivotal role in epigenetic regulations as well as in maintaining genome stability in angiosperms[44]. Compared to other eukaryotes, plants not only contain RNA Pol I, Pol II, and Pol III, they also encode Pol IV and Pol V. The former three RNA polymerases (RNAP) are mainly functional on rRNA (RNA Pol I), mRNAs, and noncoding RNAs (RNA Pol II), and tRNAs and ribosomal 5S RNAs (RNA Pol III), respectively[36,44]. Pol IV and Pol V are functionally distinct, with Pol IV being required for 24 nt siRNA biogenesis and Pol V for siRNA-mediated transcriptional gene silencing via RdDM. To further understand the evolution of siRNA metabolism, especially in the streptophyte algae, we performed gene identification and phylogenetic analyses of RDRs and five key RNA polymerases (Pol I, II, III, IV, and V).

First, we searched and identified all RDR homologs across the Archaeplastida and performed phylogenetic analyses (Fig. 3b, Fig. S12a,b). Similar to previous reports we identified two groups of RDRs (RDR1/2 and 6) in embryophytes. Previously, it was thought that RDR1/2 and RDR6 diverged from each other early in embryophyte evolution[34,36]. However, our phylogenetic analyses indicated that RDR1/2 and RDR6 originated in streptophyte algae, RDR1/2 probably after the divergence of the Klebsormidiophyceae, and RDR6 after the divergence of the Mesostigmatophyceae (Fig. 3b). Some red algae, Glaucoplantae (*C. paradoxa*), Chlorophyta, and early-diverging Streptophyta (*Mesostigma* and *Chlorokybus*) contain ancient RDR-like homologs. Surprisingly, an RDR sequence from *Mesotaenium endlicherianum* was also found in this group (Fig. 3b). We note, however, that all sequences in this group had long branches with low support values.

Specific subunits of Pol IV (*NRPD*) and V (*NRPE*) arose from the duplication of Pol II (*NRPB*) subunits and many subunits are shared by Pol II, IV, and V, but the largest subunits (*NRPB1*, *NRPD1*, and *NRPE1*) are unique to each polymerase[45]. In addition, Pol IV and V shared subunits 2, 4, 5, and 7 that are distinct from the Pol II version. To gain insight into the evolution of siRNA biogenesis across Viridiplantae, we first searched the *NRPA1/B1/C1/D1/E1* genes, which encode the largest subunit of each RNA polymerase, from all the selected species for phylogenetic analysis (Supplementary data S2 and Fig. 4a). Keeping consistent with previous reports, four groups could be clearly distinguished in our phylogenetic tree: *NRPA1/B1/C1* could be identified among all Archaeplastida[34,45]. But the non-distinguishable group of NRPD1/E1 started to appear first in bryophytes. We further examined the phylogeny of NRPD1 and NRPE1 by adding more bryophyte sequencing data. You and co-workers suggested a clear divergence of NRPD1 and NRPE1 among few fern species[34]. However, in their work, the NRPD1/E1 genes could not be confidently assigned to either the NRPD1 or the NRPE1 clade in bryophytes and lycophytes. The divergence of Pol IV and Pol V thus probably occurred in the most recent common ancestor of ferns and seed plants. The sister group of extant ferns and seed plants probably had a single Pol II derivative. NRPD1/E1 could be identified in bryophytes including hornworts, liverworts, and mosses, while no NRPD1/E1 homolog was found in the streptophyte algal genomes (Supplementary data S4). Here, our phylogenetic analyses by using maximum

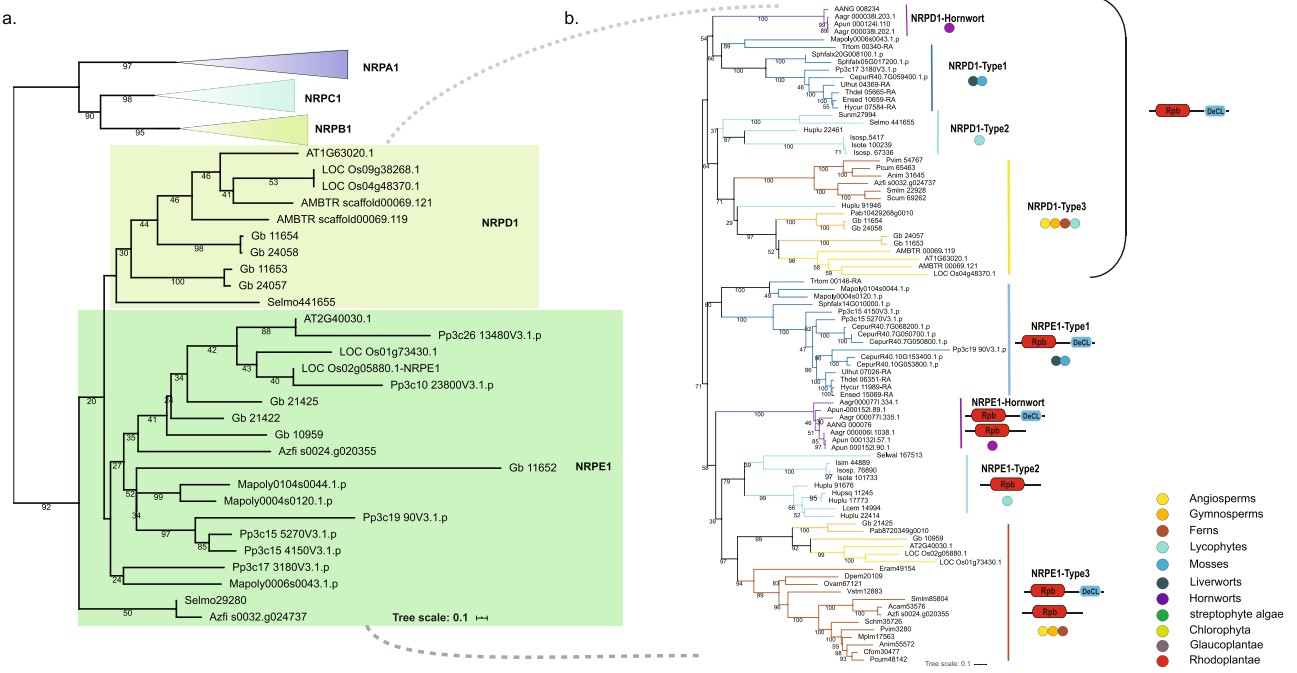

**Fig. 4 Phylogenetic trees of subunit 1 of RNA polymerase. a** A maximum likelihood tree for subunit 1 of RNA polymerase in selected Viridiplantae species. **b** A maximum likelihood tree of subunit 1 of RNA polymerase using an extended taxon sampling of bryophytes. The domain information is displayed for each group. The tree was constructed by maximum likelihood using 500 bootstrap replicates, values ≥30% are shown on internal branches. The PROTCATGTR model was predicted by Modelfinder. For species designations of sequence IDs, see Supplementary data S3.

likelihood and Mr. Bayes further revealed that the divergence of NRPD1/E1 presumably took place in the last common ancestor of embryophytes (Figs. 4b, S13, S14). The NRPD1 could be separated into four groups including a hornwort-specific type (NRPD1-hornwort), NRPD1-Type1, comprising genes from liverworts and mosses, but no lycophytes. The third group includes genes from lycophytes (NRPD1-Type2). NRPD1 genes from the lycophyte *Huperzia lucidula*, ferns, gymnosperms, and angiosperms belonged to the fourth group (NRPD1-Type 3). Similar phylogenetic types and species distributions were also observed for NRPE1, which was supported by our RAxML, Mrbayes, and IQtree analyses (Figs. 4b, S13, 14).

Next, we investigated the second largest subunit of the polymerases to evaluate the divergence of Pol IV/Pol V from Pol II. In accordance with prior findings[34,36], NRPA2/B2/C2 formed three monophyletic groups, respectively, showing that they are distinct from one another and extremely conserved across Viridiplantae (Fig. S15). NRPA2 and NRPC2 are sister genes and presumably originated through a gene duplication in the last common ancestor of Chlorophyta and Streptophyta, whereas NRPB2 presumably originated in the last common ancestor of Archaeplastida (Fig. S15). Homologs of NRPD2 were found in bryophytes, lycophytes, ferns, and seed plants, and presumably originated in the last common ancestor of the embryophytes (Fig. S15). Gene identification and phylogenetic analysis of the third subunit (NRPA3 and NRPB3) generally followed the species phylogeny, indicating that both the algal NRPA3 and NRPB3 genes formed a monophyletic group with their respective embryophyte homologs (Fig. S16). NRPB4/D4, encoding the fourth subunit of Pol II, and IV, showed high sequence similarity in Arabidopsis. Therefore, we further analyzed the origin, evolution, and differentiation of these two genes by performing a phylogenetic analysis across the Archaeplastida. The homologs of NRPB4 were detected in Chlorophyta and Streptophyta, but not in Rhodoplantae and Glaucoplantae. NRPD4 from all examined genomes of embryophytes formed a

monophyletic group separate from that of NRPB4, and only included gymnosperms and angiosperms (Fig. 5a), indicating that NRPD4 may have evolved through a gene duplication from NRPB4 in the last common ancestor of gymnosperms and angiosperms.

The phylogeny of genes for the 5th subunit of Pol II, Pol IV, and Pol V was also analyzed to evaluate the evolution of Pol IV/ Pol V from Pol II. None of the genes of streptophyte algae, bryophyte, and lycophyte species belonged to the annotated *Arabidopsis* NRPD5/E5 group, indicating that this subunit of Pol IV/V might have evolved in later-diverging angiosperms[36]; and Fig. S17. Finally, we explored the evolution of the 7th subunit of Pol I, II, III, Pol IV, and Pol V. Our phylogenetic tree depicted NRPD/E7 as a monophyletic group separating it from NRPB7 (Fig. 5b). Similarly, no NRPD/E7 homologs of streptophyte algae could be detected, suggesting that the 7th subunit of Pol IV/V probably evolved in the last common ancestor of embryophytes.

## Discussion

Small RNAs play a major role in the transcriptional (RdDM/Pol IV/V pathway) as well as post-transcriptional regulation of gene expression in eukaryotes[2,4]. The exploration of small RNA species, their functions, and particularly their origin and evolution has thus gained substantial interest over the past decade[6,10]. Despite the evolutionary importance of streptophyte algae, knowledge on small RNAs in this group of green algae is almost non-existent[34,36]. Therefore, to fill this gap, we performed a comprehensive genome-wide analyses of small RNA biosynthetic and degradation pathways, and identified the possible origin and diversification of several key genes involved in these pathways.

According to our findings, two evolutionary transitions in streptophyte algae were particularly important in relation to responses to abiotic and biotic stresses (Fig. 6): (1) The evolution of homologs of DCL-New, DCL1, AGO1/5/10, and AGO4/6/9 in the last common ancestor of Klebsormidiophyceae and all other streptophytes, which are mainly involved in responses to abiotic

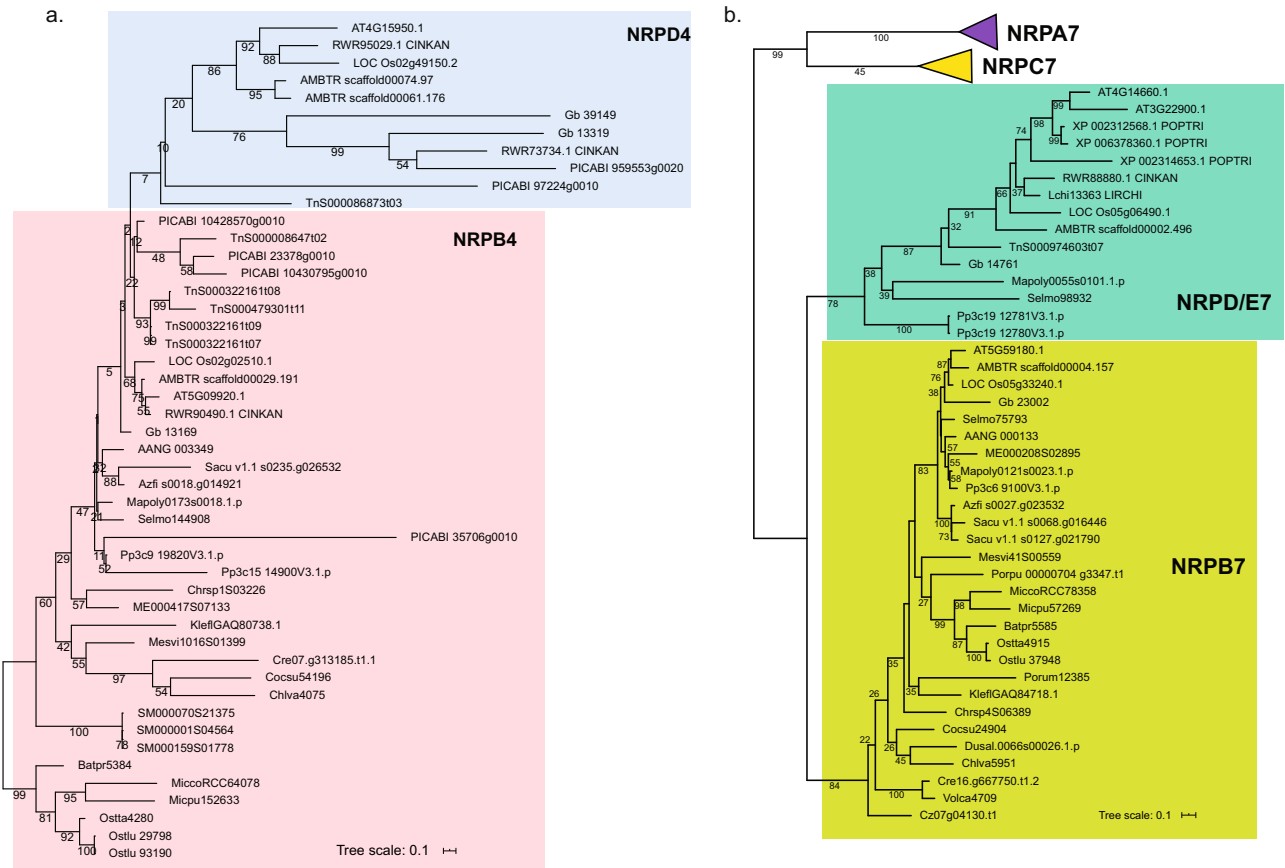

**Fig. 5 Phylogenetic analyses of subunits 4 and 7 of RNA polymerase. a** A maximum likelihood tree of subunit 4 of RNA polymerase in selected Viridiplantae species. **b** A maximum likelihood tree of subunit 7 of RNA polymerase in selected Viridiplantae species. The tree was constructed by maximum likelihood using 500 bootstrap replicates, values ≥20% are shown on internal branches. For species designations of sequence IDs, see Supplementary data S3.

stress[46], plant hormones, salt stress[47,48], antiviral defense (AGO1,10)[49,50], and regulation of HD-ZIPIII[51]; (2) Evolution of DCL 2,3,4, and AGO 2,3,7 as well as DRB1 in the last common ancestor of Zygnematophyceae and embryophytes, which likely contributed to pathogen defense, antiviral defense (e.g., AGO2/7) and antibacterial immunity[52]. In addition, we also evaluated the origin and expansion of siRNA pathway-related genes in embryophytes, particularly the NRPDs (NRPD/E1, NRPD2, NRPD/NRPE7) in the ancestor of embryophytes, and DRB4, NRPD4, NRPD5/NRPE5, and AGO-like genes in the ancestor of vascular plants (Fig. 6). Presumably, these newly expanded gene families contributed significantly to the evolution and differentiation of tissues and organs in embryophytes and to their adaptation to different terrestrial environments[19,21,36,53].

**New insights into the evolution of DCL and AGO genes using phylogenetic analyses.** We first focused on the Argonaute protein[54] and dicer-like protein (DCL) factors which are the two most critical components of the RNA silencing machinery in plants. Contrary to previous findings, we identified 5 DCL clades (excluding the algal-type DCL). Earlier, only DCL1-4 types were known[34,36,55]. Here we identified a new DICER group (DCL-New) in only a few lineages of streptophytes (from Klebsormidiophyceae to gymnosperms) and it seemed to be absent in angiosperms, suggesting a gradual loss of function of DCL-New group. Future genetic transformation experiments should provide detailed insights into functional diversification of this new DCL group. Furthermore, the presence of algal-like DCLs only in

*Mesostigma* and *Chlorokybus* (the earliest-diverging streptophyte algae), among Streptophyta, likely represents the plesiomorphic state of DCLs in Archaeapastida (Fig. 6).

We further found that DCL1 presumably originated in the last common ancestor of Klebsormidiophyceae and all other streptophytes, i.e., after the divergence of the Mesostigmatophyceae. The transition between Mesostigmatophyceae and Klebsormidiophyceae is characterized by the evolution of multicellularity in streptophytes (Fig. 1). Klebsormidiophyceae display uniseriate, unbranched, and non-polar filaments and it is possible that the origin of DCL1 (and AGO1/5/10) during this transition is related to the regulation and maintenance of coordinated cell divisions that are required for filament formation (for the role of DCL1/ AGO1 in the regulation of cell divisions in *Arabidopsis* see Trolet et al.[56].

We also gained key insights into the origin of DCL2/3/4, and found that it presumably evolved in the last common ancestor of Zygnematophyceae and embryophytes. The presence of three copies of DCL2/3/4 in the *Spirogloea muscicola* genome is attributed to the recently reported genome triplication event in this species[21]. WGDs, which are prominent in embryophytes, have been shown to play a crucial role in the diversification of small RNA pathway-related genes (siRNA pathway)[36].

Ten types of AGO genes are known in *Arabidopsis*, and are classified into four main clades among embryophytes; AGO 1/5/10, AGO 2/3/7, AGO 4/6/8/9, and AGO-like[18]. Our phylogenetic tree was consistent with previous observations[34], however, but we were also able to identify the evolutionary origin of each clade of AGO genes. Algal-type AGO genes were

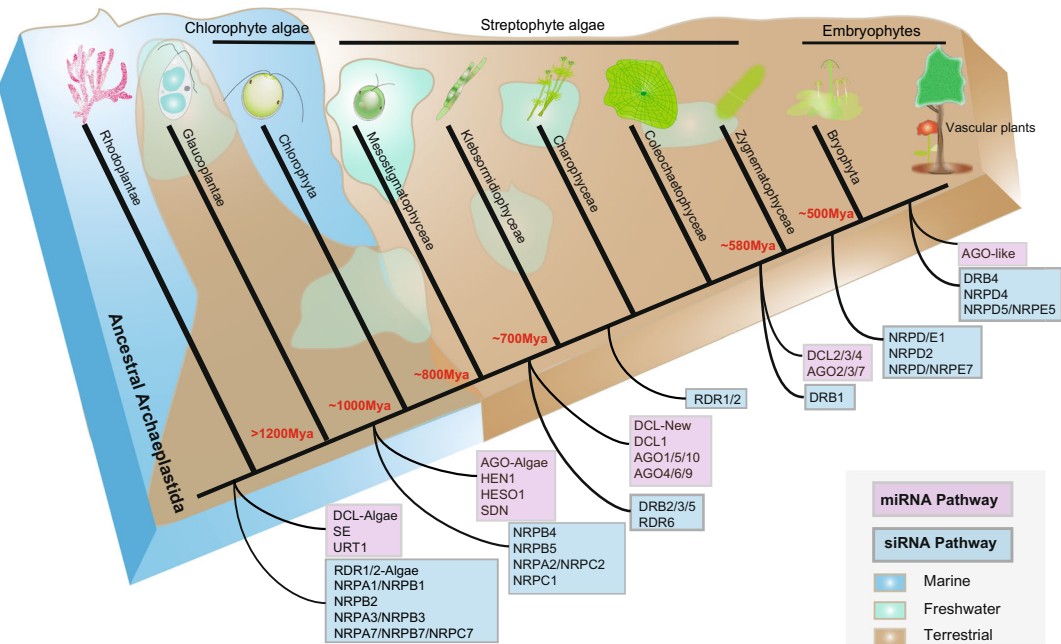

**Fig. 6 A schematic presentation of the evolution of small RNA pathway-related genes across Archaeplastida based on the phylogenetic analyses presented in Figs. 3–5 and 6 Supplementary Figs. 1–10.** Blue and purple boxes represent the siRNA and miRNA pathway-related genes, respectively. Habitat distribution of different taxa is shown in different colors: brown (subaerial/terrestrial), light green (freshwater), blue (marine). Mya (in red) indicates divergence time in million years for the next node. Further details are presented in the "Results" and "Discussion" sections.

found to be present among streptophytes only in the earliest-diverging clade (Mesostigmatophyceae). In the AGO4/6/8/9 clade, two copies from the *K. nitens* genome and from later-diverging streptophyte algae suggested that the ancestral form of AGO 4/6/8/9 genes presumably originated in streptophyte algae after the divergence of the Mesostigmatophyceae. In addition, *K. nitens*, *C. braunii*, and Zygnematophyceae genomes also encode ancestral forms of AGO 1/5/10 homologs, suggesting that both clades AGO4/6/8/9 and AGO 1/5/10 presumably originated and underwent diversification at the same time (Figs. 2b and 6). To further confirm our hypothesis, we used different methods and software, and obtained almost identical topological structure despite the relatively lower bootstrap and posterior probability values. In the IQtree result, two sequences (KleflGAQ77599 and KleflGAQ89017) of *Klebsormidium* diverged from the clade 1/5/10 and formed an individual group which was not classified in any known plant AGO clades. However, our RAxML and MrBayes trees suggested that the two AGO sequences of *Klebsormidium* belonged to the plant-like AGO (1/5/10), and similarly the two sequences of *Chlorokybus* which previously belonged to AGO-Algae (from RAxML and Bayes tress) now belonged to a clade distinct from known AGO clade (based on IQ tree) (Fig. 2b, c; Figs. S4, S5). These results indicate that the ancestral states of the plant-like AGO started to appear after the divergence of the Mesostigmatophyceae. Remarkably, our genome-wide analyses also revealed that only ferns and gymnosperms display unique AGO-like homologs, which supported previous studies that AGO-like homologs are only present in some lycophytes, ferns, and gymnosperms but are absent in the annotated genomes of angiosperms[34,36]. Based on the AU test and the tree topologies based on RAxML and MrBayes we conclude that (i), the ancestral plant-AGO evolutionarily appeared in the early streptophyte algae after divergence of Mesostigmatophyceae[30], and can be tentatively classified as plant AGO 1/5/10. Overall, the early origin and diversification of plant-type DICER and AGO among streptophyte algae could be indicative of early

adaptations of these algae to a subaerial habitat as hypothesized recently for the most part of streptophyte algal evolution[21,29,57].

**The origin and diversification of other miRNA pathway-related genes.** Our comprehensive genome-wide analyses of both biogenesis and degradation-related genes of sRNA metabolism across Archaeplastida revealed consistent results with previous studies, suggesting the presence of HEN1, HESO1, URT1, and SDN in the most recent common ancestor of embryophytes[34,36]. Although *SE* was detected as a single copy gene in algae, its multiple copies in embryophytes might suggest additional roles. Similarly, both HEN1 and HESO1 were observed to be distributed mostly in single copy form in Viridiplantae, HESO1 is involved in RNA uridylyltransferase activity, and is completely inhibited by 2′-O-methylation in *Arabidopsis*[58]. Interestingly, we also identified the possible origin of the ancestral SDN and it probably evolved after the divergence of Glaucoplantae (Fig. 6).

Double-stranded RNA-binding (DRB) protein functions in microRNA (miRNAs) biogenesis by assisting DCL1 in the accurate processing from primary miRNAs (pri-miRNAs) to miRNAs in the nucleus[39]. Consistent with the previous grouping, we also identified three major DRB groups comprising DRB2/3/5, DRB1, and DRB4[39]. Moreover, DRB3 has been shown to participate in methylation-mediated defense against gemini-viruses in *Arabidopsis*, suggesting perhaps a similar role in streptophyte algal lineages against viral attack[59]. The presence of DRB2/3/5 homologs in *K. nitens* and other streptophyte genomes followed by the diversification of DRB1 and DRB4 also suggests functional divergence across Streptophyta.

**Evolutionary insights into siRNA biogenesis-related RNA polymerases.** siRNA molecules have emerged as critical regulators in the expression and function of eukaryotic genomes. RNA polymerases (RNAP) are a class of enzymes which in vivo synthesize RNA molecules using double-stranded DNA as a template[60]. In comparison to other eukaryotes, plants contain two

additional polymerases (RNA Pol IV and Pol V), in addition to RNA Pol I, Pol II, and Pol III. We evaluated and identified the presence of these RNAP across Archaeplastida. In *Arabidopsis*, 24-nt siRNAs are the main form which conduct siRNA-directed gene silencing[20]. Their biogenesis and functionalization mainly rely on Pol IV/V and RDRs (Both RDR2 and RDR6 are reported to participate in RdDM and posttranscriptional gene silencing in *Arabidopsis*.)[20,61]. We identified two types of RDRs (RDR1/2 and RDR6) with conserved domains (additional files) which is consistent with earlier reports[61]. Earlier studies mainly focused on the functional characterization of RDRs, but there is no phylogenetic analysis for these important genes. We explored and identified the early diversification of RDR1/2 and RDR6, which occurred in the streptophyte algae after divergence of the Mesostigmatophyceae. RDR 1/2 apparently originated in the last common ancestor of Charophyceae and all other streptophytes (Fig. 6).

Although Pol II, IV, and V mutually share many subunits, the first and largest subunit (NRPB1, NRPD1, and NRPE1) is specific for each polymerase. Moreover, it has been shown that the specific subunits of Pol IV (*NRPD*) and V (*NRPE*) are formed due to duplication of Pol II (*NRPB*)[36,45]. The phylogenetic tree of NRPA1/B1/C1 was found to be consistent with the earlier studies[34]. But an indistinguishable group of NRPD1/E1 began to appear in bryophytes, suggesting that the divergence of Pol IV and Pol V probably occurred in the most recent common ancestor of ferns and seed plants. Similarly, NRPD1/E1 was identified in all bryophytes including the hornwort, liverwort, and moss genomes but was absent among streptophyte algae. This led us to hypothesize that NRPD1/E1 diverged earlier than previously thought, namely in the last common ancestor of embryophytes. Based on our observations we conclude that there was lineage-specific evolution of NRPE1 and NRPD1, and the complicated and step-wise diversification process in embryophytes might explain their morphological evolution and rapid adaptation to environmental stresses.

However, consistent with the earlier findings, we also observed the monophyly of NRPA2/B2/C2, suggesting their conservation and evolutionary uniform function as that of land plant homologs. But, the absence of the NRPD2 gene in streptophyte algae provides evidence for its origin in the most recent common ancestor of embryophytes (Fig. 6). Similar to NRPA, the NRPD4 also formed a monophyletic group, and its absence in streptophyte algae as well as in bryophytes suggested that NRPD4 might have evolved from NRPB4 in the gymnosperms, and could have participated in growth and development of early seed plants[62]. The NRPD5/E5 group also displayed similar phylogenetic trends as other RNAPs, and is suggested to have emerged in ferns and subsequent embryophyte lineages. The absence of NRPD/E7 in streptophyte algae and its monophyly also confirmed its origin in the last common ancestor of embryophytes. Further molecular and biochemical characterization of these genes in diverse streptophytes are expected to provide additional insight into their in situ function.

**Conclusions**. Previous evolutionary studies of sRNA biogenesis-related genes mainly focused on embryophytes. However, there are hardly any studies on the evolutionary history of genes involved in small RNA pathways prior to the divergence of embryophytes, i.e., during the evolution of streptophyte algae. In this study, we mapped enzymes of the miRNA and siRNA pathways on the phylogeny of streptophytes established from recent phylotranscriptomic and phylogenomic analyses. We show substantial evolutionary changes in sRNA pathway enzymes throughout streptophyte algal evolution. We identified the

occurrence of two major evolutionary transitions in streptophyte algae regarding both miRNA and siRNA pathways and hypothesized that the origin and diversification of several small RNA pathway-related genes during these evolutionary transitions might have played a significant role in responses to environmental stresses during adaptation to a progressively terrestrial habitat. During the first transition, the evolution of DCL-New, DCL1, AGO1/5/10, and AGO4/6/9 in the ancestor of Klebsormidiophyceae and all other streptophytes (Fig. 6) could be linked to responses to abiotic stress, including salinity tolerance, as well as to the evolution of multicellularity in streptophytes. During the second transition, the evolution of DCL 2,3,4, and AGO 2,3,7 as well as DRB1 in the last common ancestor of Zygnematophyceae and embryophytes (Fig. 6), suggests their possible contribution to pathogen defense and antibacterial immunity. Similarly, the duplication and diversification of DCL and AGO genes in embryophytes indicate functional refinement and diversification of the sRNA pathways. Our results on the evolution of sRNA pathway-related genes in streptophytes support recent conclusions that homologs of genes that play important roles in adaptations to the terrestrial environment in embryophytes, evolved earlier, namely in streptophyte algae. In addition, these genes also showed some diversification in different lineages of streptophyte algae after the divergence of the Mesostigmatophyceae, suggesting functional specialization. Thus, functional diversification of sRNA-related genes is not restricted to embryophytes as previously thought.

## Methods

**Data sources**. The genomes and transcriptome from 34 species were mainly used in this study, the genomes include 10 Embryophyta, 7 streptophyte algae, 12 Chlorophyta, 1 Glaucoplant, and 4 Rhodoplantae including the transcriptome data of *Coleochaete orbicularis* (Fig. 1). Besides, some selected transcriptome data from Figshare (https://figshare.com/) and 1KP were also retrieved for phylogenetic analyses (Supplementary data S2). The sequence data used in this study were mainly downloaded from the NCBI (https://www.ncbi.nlm.nih.gov/) and CNGB databases (https://db.cngb.org/). The detailed information of data sources and accession numbers are listed in Supplementary data S1.

**Homolog identification, multiple sequence alignment, and phylogenetic reconstruction**. To search and identify homologous genes of miRNA and siRNA metabolic pathways in each species, blastp (e-value $<10^{-5}$) was used with credible queries from model species *A. thaliana*, *O. sativa*, and *C. reinhardtii*. Pfam and Swissprot databases were combined to check the identified sequences. In this part, hmmsearch were used to identify domains of candidate proteins using HMM profile from the Pfam database (http://pfam.xfam.org/search#tabview=tab1) with parameter $E < 1e-5$. Then, these identified whole protein sequences were used to perform the phylogenetic analysis. For phylogeny reconstruction, first, multiple sequence alignments were processed by using MAFFT (version 7.310). Maximum-likelihood gene trees were build using RAxML (version 8.2.4) with the PROTCATGTR model with 500 replicate bootstrap values and iqtree (version 1.6.1) with the best model by automatic predistortions. The PROTCATGTR model was predicted by the Modelfinder (http://www.iqtree.org/ModelFinder/). The Bayesian trees were constructed by MrBayes (version 3.2.6) using the GTR-GAMMA evolutionary model with six Markov chains until the average standard deviation of split frequencies was <0.05 (AGO tree: 600,000 generations, DCL, NRPDE1, RDR tree with 500,000 generations). The final trees were drawn by iTOL (https://itol.embl.de/). The protein alignments were generated by Genious (version 2020 2.4), and ESPript (version 3.0).

**Approximately unbiased (AU) test**. The AU test was conducted by using CONSEL (v0.20)[63]. First, we used PhyML to produce the log-likelihood of site-pattern for each tree. Then we used the multi-scale bootstrap file as input in the CONSEL makermt program to calculate the p-values of each tree. Finally, the output rmt file of makermt was used as input for CONSEL's main program consel to calculate various p-values to assess the reliability. The p-values obtained by consel AU were considered as the main result.

**Gene domain comparison**. The Pfam was used to identify the domains of candidate sequences. Motifs were identified in the web of Multiple Em for Motif Elicitation 5.0.5 (http://meme-suite.org/) with default parameters.

**Reporting summary**. Further information on research design is available in the Nature Research Reporting Summary linked to this article.

## Data availability

The detailed information of data sources and the accession numbers are provided in the Supplementary Data 1.

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

## Acknowledgements
This work was supported by the Shenzhen Municipal Government of China (grant numbers No. JCYJ20151015162041454) and the Guangdong Provincial Key Laboratory of Genome Read and Write (Grant No. 2017B030301011). This work is part of the 10KP project (https://db.cngb.org/10kp/)[64]. This work is also supported by China National GeneBank (CNGB; https://www.cngb.org/).

## Author contributions
H.L. and S.K.S. conceived, designed, and supervised the project. M.M., M.P., and H.L. provided resources and materials. S.W., H.Liang., S.K.S., H.W., L.L., Y.X., and D.N.S. analyzed the data. S.K.S., S.W., and M.M. wrote the paper. M.M., S.K.S., S.W., D.N.S., H.Liang., and M.P. revised the manuscript. All the authors read and revised the final version of the manuscript.

## Competing interests
The authors declare no competing interests.
