## [Peer Review File · Communications Biology]

Reviewers' comments:

Reviewer #1 (Remarks to the Author):

Summary of the manuscript

The authors have traced the algal origin of major land plant sRNA pathway factors using phylogenetic methods. Specifically, the appearance and diversification of DCL, AGO, RDR, RNA pol IV/V and auxiliary sRNA factors were studied in Streptophyte algae, the modern-day closest ancestors of land plants. The authors claim to pinpoint the appearance of individual AGO, DCL and RDR gene families, and propose that two transitions took place in the evolution of Streptophyte sRNA function; (1) occurring after the divergence of Mesostigmatophyceae and (2) taking place in the common ancestor of Zygnematophyceae and Embryophyta. The authors propose that this brought forward new functionalities helping adaptation to life on land: (1) multicellularity & response to abiotic stress and (2) defence against pathogenic microorganisms. Some expanded gene families were found in the Streptophytes, such as AGO1/5/10, for which up to 3 genes exist in Klebsormidiophyceae and Zygnematophyceae. This indicates that functional specialization of sRNA pathways enzymes is not exclusive to embryophytes.

Overall impression of the study

The research question is interesting and relevant. Given the important roles that sRNAs play in present-day land plants, investigations on sRNA processes in Streptophyte algae is likely to shed light on some of the mechanisms that allowed plants to conquer land.

The study design is sound and the figures and tables are nicely organized and visually good-looking. The findings are novel and also exciting for researchers in the field of sRNAs and plant/algal evolution.

1. The overall positive impression is reduced by small mistakes that could have been avoided by more careful proofreading (e.g., checking the references to figures/tables and literature). I also think the authors should have done their fact-checking more thoroughly and been more careful with word choice. As an example, it is not correct to refer to DCL and AGO as "transcription factors" (L129), as this term describes a different class of gene regulatory proteins.

2. The conclusions regarding the presence/absence of DCL, AGO and RDR proteins in Streptophyta are interesting and reasonable. For example, Mesostigmatophyceae were found to have only algal-like DCL and AGO, while later branching streptophytes had embryophyte-type AGO and DCL, which makes sense in light of the increasing "complexity" of the latter algae. However, several clades in the phylogenies are supported by very low bootstrap values. With such low support for major AGO, DCL and RDR clades, I am concerned that the conclusions are not standing on solid ground. As an example, the relationship between AGO groups cannot be resolved (bootstrap <50%), which means it is difficult to firmly assign the Streptophyte protein homologs to specific clades. (Same for DCLs and RDR6.) I find the author's conclusion about the two evolutionary transitions very attractive, but not convincing. To strengthen their findings, major clades in their trees need stronger support.

Specific comments

L31: there are 34 algae/plants in Table 1 and Fig. 2.

L69: "Alternatively" is not fitting here. Perhaps "Subsequently" works better?

L75: AGO1 does not produce miRNAs, it associates with them to get guided to a target RNA.

L85: "cleavage" is the wrong term. I would recommend "by the activity of RNA polymerase IV".

L129: DCL and AGO are not transcription factors. It is enough to say "factor" or "protein factor".

L148: ref. 35 should not be cited here. I cannot find any info about DCL1 in it.

L159: this info about the four AGO clades (AGO1/5/10, AGO2/3/7, AGO4/6/8/9, AGO-like) is not

mentioned in ref. 6. Suggestion: use ref. 18 (Vaucheret 2008) instead.

L181-184: lycophytes are found as well in the AGO4-like clade. And, on what base do the authors expect this clade to be present in bryophytes? Could it not have evolved after the bryophyte divergence?

L187: HEN1, HESO1, URT1 or SDN should be in italics, if genes (L188).

L189: is ref. 35 correct here? There is no info about HEN1, HESO1, URT1 or SDN in it.

Orthologs with which genes? If not explained, I suggest deleting the part about orthology from the sentence.

L205: "Chorophyta" should be "Chlorophyta" ^{SEP}

L223-226: please add a reference for the statement that many Chlorophyta have miRNA.

L245: I don't see ref. 35 as relevant here either.

L246-247: it is the other way around; RDR1/2 after the divergence of Klebsormidiophyceae and RDR6 after the divergence of Mesostigmatophyceae (Figs 4b, 7).

L313: sRNAs also play major roles in transcriptional regulation (RdDM/Pol IV/V pathway).

L326: although AGO10 is involved in defence against viruses, so are several other Arabidopsis AGOs, and AGO1 and AGO2 have the major antiviral activity (see e.g., Carbonell & Carrington review, DOI: 10.1016/j.pbi.2015.06.013).

L328-329: along the same line, antiviral defence (AGO2/7) could be added here (alongside pathogen defence and antibacterial immunity).

L333: "genes" should be "gene families".

L489-90: Table S1 lists 34 species, whereof 10 Embryophyta and 7 Streptophyta.

L494: should be Table S2, not S1?

Methods: to generate the phylogenies, were whole protein sequences used or specific domains? (E.g., for DCL and AGO, were the characteristic RNase III respectively PIWI domain used?) Please add this information to the Methods section and/or include in each figure legend.

Fig. 2: I cannot find the reference to this figure in the main text.

Fig. 3: there are abbreviations which are not included in Tables S1/S2. Please recheck the abbreviations: correct if wrong and add to the table(s) if correct.

Fig. 7: HEN1 appears twice. I think it should be deleted from the blue box at >1200Mya.

Reviewer #2 (Remarks to the Author):

In this study, the authors identified genes involved in the small RNA pathway in Viridiplantae (data from public databases) and did a thorough phylogenetic analysis of many important genes. These results suggested that Viridiplantae started to evolve plant-like miRNA biogenesis and degradation after the divergence of the Mesostigmatophyceae in the streptophyte algae. The origin and diversification of these genes among streptophyte algae agree with previous data. This suggested progressive adaptations of streptophyte algae to a subaerial environment.

However, there are still some vital flaws preventing this manuscript from satisfying me.

1. The figures do not meet the standard for publication. The fonts should be appropriate and similar within one figure; the size of the figure should meet the requirements of the journal; most figures lack proper labels, making them difficult to understand (i.e. branches or taxa names can be colored to highlight genes or species or groups mentioned in the main text). Especially, the content of figure 1 can be easily found in previously published reviews on small RNA in plants. There is no new information in this figure. Besides, figure 2 is not mentioned in the main text.

2. The discussion part mainly talked about the results again. It should not be an emphasis on the backgrounds and results mentioned above, but interpret the results and induce critical thoughts,

hypotheses, and perspectives. For example, the authors may talk about the differences between miRNAs in *C. reinhardtii* and land plants on the form, biogenesis, and degradation.

Major points:

1. line 103-105. "streptophyte algae" should be designated as the latest common ancestor of streptophytes and embryophytes. Streptophyte algae are the current species but not the ancient ones. Besides, the term "the common ancestor" in the whole manuscript should be "the latest common ancestor". The authors should be careful when using terms.
2. line 273-274. The authors claimed a clear divergence of NRPD1 and NRPE1 based on the phylogenetic tree. However, the bootstrap support value is too low (only 37) to be confident drawing this conclusion. Also, the subclades in for NRPD1/E1 were dependent only on low bootstrap values.

Minor points:

1. line 116. "MIR" should be italic.
2. line 123. "phylogenomic" should be "phylogenetic". The authors didn't incorporate genomic information in this study.
3. line 129. DCL and AGO genes do not encode transcription factors. It also happened in line 339.
4. line 138. How do the authors define "genuine" in evolution?
5. line 159. The "AGO-like" sometimes were mentioned as "AGO4-like". These should be consistent throughout the manuscript.
6. line 175. This sentence is ambiguous.
7. line 182-183. the angiosperms are still labeled in figure 3b.

Reviewer #3 (Remarks to the Author):

This manuscript demonstrated a genome-wide analysis of small RNA biogenesis genes in 33 algal and plants. This work is critical and providing more clear knowledge in the evolution of post-transcriptional gene silencing (PTGS). I have several essential comments and suggestions for improving this manuscript. If the authors addressed these suggestions, readers would easily understand the evolution of PTGS, which is the author wants to demonstrate.

Major comments

1. All the phylogenetic trees in figures need to be further interpretant and modified. For instance, Page 7, lines 146-148. "DCL1 seems to be separated from other DCL clades although bootstrap support values for all DCL clades were generally low (Fig. 3a)". To be honest, Figure 3a is hard to easy to demonstrate the author's meaning that likes the author presented in the manuscript. These problems show the entire manuscript. The complicated phylogenetic trees with small gene IDs are thorny for readers to find the data that interpretant in the manuscript.
2. In the trees, gene IDs with an abbreviation is hard to convert to species.
3. The authors identified 33 species of PTGS genes from the database and made some interesting and important conclusions based on the phylogenetic trees. However, interpretations lack experimental or other omics data support. For instance, miRNA, siRNA, and degradome profiles might provide additional data to support the conclusions.
4. The conserved functional domains or residues for each PTGS gene between species by

alignment need to be demonstrated.

Minor comment:

1. Page 6, lin129. "DCL and AGO are two key transcription factors" should change to "DCL and AGO are two key factors."
2. Typing errors in the manuscripts. Ex. Page 16, ln 346; Page 24, ln 548; Page 26, ln 595. Please check whole manuscripts carefully and correct them.
3. In Figure 3, more explanations about the similarities and differences between AGO4-like and AGO4/6/9 clades are needed.

Response to reviewer's comments

Reviewer #1

Summary of the manuscript

The authors have traced the algal origin of major land plant sRNA pathway factors using phylogenetic methods. Specifically, the appearance and diversification of DCL, AGO, RDR, RNA pol IV/V and auxiliary sRNA factors were studied in Streptophyte algae, the modern-day closest ancestors of land plants. The authors claim to pinpoint the appearance of individual AGO, DCL and RDR gene families, and propose that two transitions took place in the evolution of Streptophyte sRNA function; (1) occurring after the divergence of Mesostigmatophyceae and (2) taking place in the common ancestor of Zygnematophyceae and Embryophyta. The authors propose that this brought forward new functionalities helping adaptation to life on land: (1) multicellularity & response to abiotic stress and (2) defence against pathogenic microorganisms. Some expanded gene families were found in the Streptophytes, such as AGO1/5/10, for which up to 3 genes exist in Klebsormidiophyceae and Zygnematophyceae. This indicates that functional specialization of sRNA pathway enzymes is not exclusive to embryophytes.

Overall impression of the study

The research question is interesting and relevant. Given the important roles that sRNAs play in present-day land plants, investigations on sRNA processes in Streptophyte algae is likely to shed light on some of the mechanisms that allowed plants to conquer land.

The study design is sound and the figures and tables are nicely organized and visually good-looking. The findings are novel and also exciting for researchers in the field of sRNAs and plant/algal evolution.

1. The overall positive impression is reduced by small mistakes that could have been avoided by more careful proofreading (e.g., checking the references to figures/tables and literature). I also think the authors should have done their fact-checking more thoroughly and been more careful with word choice. As an example, it is not correct to refer to DCL and AGO as “transcription factors” (L129), as this term describes a different class of gene regulatory proteins.

Response: Thank you for pointing out the mistake. We have modified the sentence as per the suggested corrections in the result section “DCL and AGO are the two key factors regulating sRNA metabolism”.

2. The conclusions regarding the presence/absence of DCL, AGO and RDR proteins in Streptophyta are interesting and reasonable. For example, Mesostigmatophyceae were found to have only algal-like DCL and AGO, while later branching streptophytes had embryophyte-type AGO and DCL, which makes sense in light of the increasing “complexity” of the latter algae. However, several clades in the phylogenies are supported by very low bootstrap values. With such low support for major AGO, DCL and RDR clades, I am concerned that the conclusions are not standing on solid ground. As an example, the relationship between AGO groups cannot be resolved (bootstrap <50%), which means it is difficult to firmly assign the Streptophyte protein homologs to specific clades. (Same for DCLs and RDR6.) I find the author’s conclusion about the two evolutionary transitions very attractive, but not convincing. To strengthen their findings, major clades in their trees need stronger support.

Response: Thank you for raising this important point. We have tried different methods, including the Maximum Likelihood and Bayes; different predicted models and different software including RAxML, MrBayes and IQtree, and even added more sequences into each clade with increased sampling (1KP dataset) to perform the phylogenetic analysis on the AGO, DCL and RDR. However, the bootstrap at the key nodes from all the different phylogenetic trees still displayed low bootstrap values. Thus, the low bootstrap values were not caused by unsuitable methods or models of evolution but are rather due to the difference in the intrinsic sequence variation in the ancestral plant-like AGO with respect to Chlorophyta, and the mature AGO (of embryophyte land plants). This could be the

possible reason for the lower bootstrap for DCL and RDR as well. For example, we performed the AGO phylogenetic analysis by using RAxML, MrBayes and IQtree. The RAxML and MrBayes showed same topological structure even when the bootstrap was relatively low; however, in the IQtree result, two sequences (KleflGAQ77599 and KleflGAQ89017) of *Klebsormidium* diverged from the clade 1/5/10 and formed an individual group which were not classified in any known plant AGO clades. Likewise, our RAxML tree and MrBayes tree supported the two sequences of *Chlorokybus* belonging to the AGO-algae clade. However, the two sequences in the IQtree diverged from the AGO-algae clade. These results might suggest that the ancestral states of plant-like AGO started to appear at these evolutionary nodes (i.e., *Chlorokybus* and *Klebsormidium*), the ancestral plant-like AGO sequences displayed a preliminary difference (intermediate state between ancestral AGO algae and mature embryophyte AGO) between AGO-algae clade, however, it also displayed significant difference with true (mature) plant AGOs that were evolved >500 mya from their ancestral plant-like AGOs (Ex: AGO from *Klebsormidium* to *Arabidopsis*). Thus, our result provide evidence that the plant-like AGO or the ancestral plant-like AGO started to appear and evolve at the evolutionary nodes of *Chlorokybus* and *Klebsormidium*. These ancestral plant-like AGO is difficult to be classified into a mature plant-AGO clade since their sequences appeared to be primordial chaos (started to deviate from AGO-algae and gradually became closer to that of the plant AGO). Thus, the evolutionary differences in intrinsic sequences among AGO-algae, ancestral plant-like AGO and plant AGOs caused the low bootstrap value, and we strongly believe that the obtained result will not affect our final conclusion that the plant-AGO or ancestral plant-AGO evolutionary appeared at the early streptophyta algae (i.e., *Chlorokybus* and *Klebsormidium*) (or after *Mesostigmatophyceae*).

Because of the scarcity of genome data from the Streptophyta algae, the conclusions are tentative. Further availability of more genomic sequences would certainly provide more compelling evidences.

The next question was which AGO clade did the earliest ancestral plant-like AGO might belong?

RAXML, MrBayes and IQ tree displayed different topologies. Though, initially we could not confirm which tree is completely correct or wrong, but all the trees clearly depicted the possible classification of ancestral plant-like AGO (fig 2c). We thus performed an AU test (see methods), and sequence identity comparisons to check which phylogeny is the most reliable. The AU test demonstrated that the phylogeny using of RAXML and MrBayes were more reliable than that of the IQ tree (see the p-value in Fig2c). Additionally, the identity of two ancestral plant-like AGO sequences (KleflGAQ77599 and KleflGAQ89017) was more similar to plant AGO 1/5/10. Moreover, the two *Chlorokybus* sequences (Chrsp23S03751 and Chrsp6S07414) were closer to the AGO-algae sequences (i.e., *Chlamydomonas reinhardtii*) than the other streptophyte sequences (i.e., KleflGAQ77599) (although the two sequences might appear to be distinct with AGO-algae, and appear similar to plant-like AGO). Thus, we derived two conclusions from our analyses: (1), the plant-AGO or ancestral plant-AGO evolved in the early-diverging streptophyte algae; (2) the ancestral plant-AGO sequences can be mainly classified into the plant AGO 1/5/10.

Likewise, the low bootstrap values in the Dicer and RDR phylogenetic trees also resulted from the ancestral plant-like Dicer/RDR rather than the phylogenetic method or evolutionary model used or a sampling bias. The best model was predicted by the Modelfinder (<http://www.iqtree.org/ModelFinder/>). Our RAXML, MrBayes and IQ tree all displayed similar phylogeny.

Specific comments

L31: there are 34 algae/plants in Table 1 and Fig. 2.

Response: Corrected “We used genome and transcriptome data of 34 algal and plant species”

L69: “Alternatively” is not fitting here. Perhaps “Subsequently” works better?

Response: Corrected “Subsequently, the nascent miRNA/miRNA* duplex generated by DCLs is methylated by the small RNA methyltransferase HUA Enhancer 1 (HEN1)”

L75: AGO1 does not produce miRNAs, it associates with them to get guided to a target RNA.

Response: Corrected “*Arabidopsis* has 10 AGO proteins, and previous studies showed that AGO1 was the major effector protein guiding the production of miRNAs”

L85: “cleavage” is the wrong term. I would recommend “by the activity of RNA polymerase IV”.

Response: Corrected “Single-stranded RNAs (ssRNA) are generated from RdDM target loci by the activity of RNA polymerase IV (Pol IV)”

L129: DCL and AGO are not transcription factors. It is enough to say “factor” or “protein factor”.

Response: Corrected “DCL and AGO are the two key factors regulating sRNA metabolism, therefore, we firstly identified the homologues of DCL and AGO from the published Archaeplastida genomes”

L148: ref. 35 should not be cited here. I cannot find any info about DCL1 in it.

Response: The reference has been removed.

L159: this info about the four AGO clades (AGO1/5/10, AGO2/3/7, AGO4/6/8/9, AGO-like) is not mentioned in ref. 6. Suggestion: use ref. 18 (Vaucheret 2008) instead.

Response: The suggested reference has been added now.

L181-184: lycophytes are found as well in the AGO4-like clade. And, on what base do the authors expect this clade to be present in bryophytes? Could it not have evolved after the bryophyte divergence?

Response: Thank you for pointing out the mistake. We have added the correct details in the Fig. 2b. “Genome-wide analyses also revealed that the long-branch AGO-like clade

only occurred in lycophytes, ferns and gymnosperms (Fig. 2b; Figs. S5, S6), its absence in bryophytes and angiosperms currently remains elusive.”

L187: HEN1, HESO1, URT1 or SDN should be in italics, if genes (L188).

Response: The gene names are italicized throughout the manuscript

L189: is ref. 35 correct here? There is no info about HEN1, HESO1, URT1 or SDN in it. Orthologs with which genes? If not explained, I suggest deleting the part about orthology from the sentence.

Response: The wrong reference has been removed now. The ortholog details are added now Supplementary Table S3.

“A previous study indicated that HEN1, HESO1, URT1, and SDN existed in the most recent common ancestor of embryophytes as single-copy genes, raising the possibility that they are orthologues in Viridiplantae (Supplementary Table S3)^{33, 35}.”

L205: “Chorophyta” should be “Chlorophyta”.

Response: Corrected

L223-226: please add a reference for the statement that many Chlorophyta have miRNA.

Response: The following references were added.

Molnár, A., Schwach, F., Studholme, D. J., Thuenemann, E. C., & Baulcombe, D. C. (2007). miRNAs control gene expression in the single-cell alga *Chlamydomonas reinhardtii*. *Nature*, 447(7148), 1126-1129.

Li, J., Wu, Y., & Qi, Y. (2014). MicroRNAs in a multicellular green alga *Volvox carteri*. *Science China Life Sciences*, 57(1), 36-45.

L245: I don't see ref. 35 as relevant here either.

Response: The reference has been removed now.

L246-247: it is the other way around; RDR1/2 after the divergence of Klebsormidiophyceae and RDR6 after the divergence of Mesostigmatophyceae (Figs 4b, 7).

Response: The suggested correction has been made.

L313: sRNAs also play major roles in transcriptional regulation (RdDM/Pol IV/V pathway).

Response: The suggested information is added now in the discussion. “Small RNAs play a major role in the transcriptional (RdDM/Pol IV/V pathway) as well as post-transcriptional regulation of gene expression in eukaryotes ^{2,4}.”

L326: although AGO10 is involved in defence against viruses, so are several other Arabidopsis AGOs, and AGO1 and AGO2 have the major antiviral activity (see e.g., Carbonell & Carrington review, DOI: 10.1016/j.pbi.2015.06.013).

Response: The suggested information is added now with suitable reference.

“(1) The evolution of homologs of DCL-New, DCL1, AGO1/5/10 and AGO4/6/9 in the last common ancestor of Klebsormidiophyceae and all other streptophytes, which are mainly involved in responses to abiotic stress ⁴⁵, plant hormones, salt stress ^{46,47}, antiviral defense (AGO1,2,10) ^{48,49} and regulation of HD-ZIPIII ⁵⁰.”

L328-329: along the same line, antiviral defence (AGO2/7) could be added here (alongside pathogen defence and antibacterial immunity).

Response: The information is added now.

“(2) Evolution of DCL 2,3,4, and AGO 2,3,7 as well as DRB1 in the last common ancestor of Zygnematophyceae and embryophytes, which likely contributed to pathogen defense, antiviral defence (e.g., AGO2/7) and antibacterial immunity ⁵¹.”

L333: “genes” should be “gene families”.

Response: corrected

L489-90: Table S1 lists 34 species, whereof 10 Embryophyta and 7 Streptophyta.

Response: The suggested information is added now.

“The genomes and transcriptome from 34 species were mainly used in this study, the genomes include 10 Embryophyta, 7 streptophyte algae, 12 Chlorophyta, 1 Glaucoplant, and 4 Rhodoplantae including the transcriptome data of *Coleochaete orbicularis* (Fig. 1).”

L494: should be Table S2, not S1?

Response: Corrected

Methods: to generate the phylogenies, were whole protein sequences used or specific domains? (E.g., for DCL and AGO, were the characteristic RNase III respectively PIWI domain used?) Please add this information to the Methods section and/or include in each figure legend.

Response: To generate the phylogenies, whole protein sequences were used. The details are now included in the methods section.

Fig. 2: I cannot find the reference to this figure in the main text.

Response: It is mentioned in line 351

Fig. 3: there are abbreviations which are not included in Tables S1/S2. Please recheck the abbreviations: correct if wrong and add to the table(s) if correct.

Response: all the abbreviations are rechecked, and missing ones are added now.

Fig. 7: HEN1 appears twice. I think it should be deleted from the blue box at >1200Mya.

Response: Fig. 6 is modified as per the suggestion.

Reviewer #2 (Remarks to the Author):

In this study, the authors identified genes involved in the small RNA pathway in Viridiplantae (data from public databases) and did a thorough phylogenetic analysis of many important genes. These results suggested that Viridiplantae started to evolve plant-like miRNA biogenesis and degradation after the divergence of the Mesostigmatophyceae in the streptophyte algae. The origin and diversification of these

genes among streptophyte algae agree with previous data. This suggested progressive adaptations of streptophyte algae to a subaerial environment.

However, there are still some vital flaws preventing this manuscript from satisfying me.

1. The figures do not meet the standard for publication. The fonts should be appropriate and similar within one figure; the size of the figure should meet the requirements of the journal; most figures lack proper labels, making them difficult to understand (i.e. branches or taxa names can be colored to highlight genes or species or groups mentioned in the main text). Especially, the content of figure 1 can be easily found in previously published reviews on small RNA in plants. There is no new information in this figure. Besides, figure 2 is not mentioned in the main text.

Response: We have now thoroughly revised the figures (Main Figs. 2,3,4,6, and Supplementary Figures S1,S2,S3,S4,S5,S6,S13,S14) as per your suggestions and journal standards. We have also moved Fig. 1 to Supplementary Figure S1.

2. The discussion part mainly talked about the results again. It should not be an emphasis on the backgrounds and results mentioned above, but interpret the results and induce critical thoughts, hypotheses, and perspectives. For example, the authors may talk about the differences between miRNAs in *C. reinhardtii* and land plants on the form, biogenesis, and degradation.

Response: We have deleted the repetitive sentences and added new discussion. The differences between miRNAs in *C. reinhardtii* and land plants on the form, biogenesis, and degradation is described briefly in the discussion section as follows: “In the chlorophyte green alga (*C. reinhardtii*), miRNA-induced gene silencing is very similar to that of higher land plants, particularly in terms of the activation of the RNA-induced silencing complex and mRNA targeting³⁹. In contrast to animals, plant miRNAs require near-perfect complementarity to direct the cleavage of target mRNAs. In plants, *MIR* genes are embedded within the noncoding sequences between protein-coding genes, while in *Chlamydomonas*, *MIR* genes are embedded within the introns of protein-coding

genes and transcribed by Pol II, similar to animal MIR (plesiomorphy)⁴⁰. This difference in the miRNA machinery between green algae and embryophytes could be due to the difference in habitat, various other environmental adaptations or structural and developmental complexity.”

Major points:

1. line 103-105. "streptophyte algae" should be designated as the latest common ancestor of streptophytes and embryophytes. Streptophyte algae are the current species but not the ancient ones. Besides, the term "the common ancestor" in the whole manuscript should be "the latest common ancestor". The authors should be careful when using terms.

Response: The suggested correction has been implemented now. We have replaced "the common ancestor" with “the last common ancestor” wherever necessary.

2. line 273-274. The authors claimed a clear divergence of NRPD1 and NRPE1 based on the phylogenetic tree. However, the bootstrap support value is too low (only 37) to be confident drawing this conclusion. Also, the subclades in for NRPD1/E1 were dependent only on low bootstrap values.

Response: Not only our phylogenetic tree supported the divergence of NRPD1 and NRPE1, but some published studies supported the classification between NRPD1 and NRPE1. Please refer reference 33 and 35. We have tried different methods, including the Maximum Likelihood and Bayes; different models (The best model was predicted by the Modelfinder (<http://www.iqtree.org/ModelFinder/>), and different software including RAxML, MrBayes and IQtree, and even added more sequences (1KP dataset) into each clade with increased sampling to perform the phylogenetic analysis on the NRPD1/E1. The bootstrap value of RAxML at the key divergent node increased to 71 % with increased sampling. In addition, other methods MrBayes and IQtree phylogeny also supported the divergence of NRPD1 and NRPE1, as well as the subclades in for NRPD1/E1.

The low bootstrap values at some nodes were not caused by the unsuitable methods rather due to the unremarkable sequence variation in the ancestral plant between NRPD1 and E1 as well as internal subclades among NRPD1-3 and NRPE1-3 (e.g., Bryophytes including the moss, hornwort, and liverworts) (Fig. 4a,b). These ancestral states of NRPD1/E1 of early Embryophyta just started to appear and diverged from NRPA/B/C. To some extent, the similarity between ancestral states of NRPD1/E1 in early Embryophyta caused the relatively low bootstrap values. Thus, we performed different phylogenetic methods and increased sampling. All the results still keep same topology and support divergence of NRPD1 and NRPE1 as well as subclades in for NRPD1-3/E3.

Minor points:

1. line 116. "MIR" should be italic.

Response: Corrected

2. line 123. "phylogenomic" should be "phylogenetic". The authors didn't incorporate genomic information in this study.

Response: corrected

3. line 129. DCL and AGO genes do not encode transcription factors. It also happened in line 339.

Response: The suggested correction has been implemented now.

4. line 138. How do the authors define "genuine" in evolution?

Response: We revised the confusing sentence. "The phylogenetic tree also indicated that DCL1 and other DCLs presumably evolved after the divergence of the Mesostigmatophyceae in the last common ancestor of Klebsormidiophyceae and all other Streptophyta (Fig. 2a)."

5. line 159. The "AGO-like" sometimes were mentioned as "AGO4-like". These should be consistent throughout the manuscript.

Response: Changed to AGO-like for consistency

6. line 175. This sentence is ambiguous.

Response: The suggested correction has been implemented now.

“Two copies of AGO genes from the *K. nitens* genome as well as AGO genes from *Chara braunii* and *Mesotaenium endlicherianum* were identified in this group, indicating that the ancestral form of the AGO 4/6/8/9 genes originated after the divergence of the Mesostigmatophyceae but before divergence of Klebsormidiophyceae (Fig. 2b, Figs. S5, S6). The same was true for the AGO 1/5/10 clade, in which three AGO genes from *K. nitens*, one gene from *C. braunii*, two genes from *M. endlicherianum*, and three genes of *Spirogloea muscicola* were found.”

7. line 182-183. the angiosperms are still labeled in figure 3b.

Response: Angiosperms are removed from the label

Reviewer #3 (Remarks to the Author):

This manuscript demonstrated a genome-wide analysis of small RNA biogenesis genes in 33 algal and plants. This work is critical and providing more clear knowledge in the evolution of post-transcriptional gene silencing (PTGS). I have several essential comments and suggestions for improving this manuscript. If the authors addressed these suggestions, readers would easily understand the evolution of PTGS, which is the author wants to demonstrate.

Major comments

1. All the phylogenetic trees in figures need to be further interpretant and modified. For instance, Page 7, lines 146-148. “DCL1 seems to be separated from other DCL clades although bootstrap support values for all DCL clades were generally low (Fig. 3a)”. To be honest, Figure 3a is hard to easy to demonstrate the author’s meaning that likes the author presented in the manuscript. These problems show the entire manuscript. The complicated phylogenetic trees with small gene IDs are thorny for readers to find the data that interpretant in the manuscript.

Response: We have now thoroughly revised the figure as per your suggestions and journal standards.

2. In the trees, gene IDs with an abbreviation is hard to convert to species.

Response: The suggested correction has been implemented now. Now we used simplified IDs, and different colors to make the reading clearer.

3. The authors identified 33 species of PTGS genes from the database and made some interesting and important conclusions based on the phylogenetic trees. However, interpretations lack experimental or other omics data support. For instance, miRNA, siRNA, and degradome profiles might provide additional data to support the conclusions

Response: Thank you for the suggestion. However, the major goal of our study was to trace the algal origin of major embryophyte sRNA pathway factors using phylogenetic methods. Specifically, the appearance and diversification of DCL, AGO, RDR, RNA pol IV/V and auxiliary sRNA factors in streptophyte algae, and their possible origin. The suggested addition of miRNA, siRNA, and degradome profiles would be important to gain functional insights, which is beyond the scope of our present study. But we will certainly consider your suggestion for our ongoing research and upcoming manuscripts.

4. The conserved functional domains or residues for each PTGS gene between species by alignment need to be demonstrated.

Response: The conserved functional domain information is now added in Additional files.

Minor comment:

1. Page 6, lin129. “DCL and AGO are two key transcription factors” should change to “DCL and AGO are two key factors.”

Response: corrected

2. Typing errors in the manuscripts. Ex. Page 16, ln 346; Page 24, ln 548; Page 26, ln 595. Please check whole manuscripts carefully and correct them.

Response: corrected

3. In Figure 3, more explanations about the similarities and differences between AGO4-like and AGO4/6/9 clades are needed.

Response: AGO4-like is mentioned as AGO-like in the revised version. Following explanations are added

“AGO genes in embryophytes can be classified into four main clades based on our maximum likelihood and Bayesian trees, AGO 1/5/10, AGO 2/3/7, AGO 4/6/8/9 and AGO-like 18 (Fig. 2b, Figs. S5, S6).”...

.. “The AGO 4/6/8/9 clades (represented as a single group hereafter) were found to be more closely related to the algal type AGOs than the other AGO clades. Two copies of AGO genes from the *K. nitens* genome as well as AGO genes from *Chara braunii* and *Mesotaenium endlicherianum* were identified in this group, indicating that the ancestral form of the AGO 4/6/8/9 genes originated after the divergence of the Mesostigmatophyceae but before divergence of Klebsormidiophyceae (Fig. 2b, Figs. S5, S6). The same was true for the AGO 1/5/10 clade, in which three AGO genes from *K. nitens*, one gene from *C. braunii*, two genes from *M. endlicherianum*, and three genes of *Spirogloea muscicola* were found.”

....

“Genome-wide analyses also revealed that the long-branch AGO-like clade only occurred in lycophytes, ferns and gymnosperms (Fig. 2b; Figs. S5, S6), its absence in bryophytes and angiosperms currently remains elusive.”

REVIEWERS' COMMENTS:

Reviewer #1 (Remarks to the Author):

The authors have improved the manuscript to a satisfactory level. Overall, I find that the authors have addressed the referees' comments in their manuscript and rebuttal letter. A few exceptions are listed below.

1. Thank you. I have no further comments.

2. I appreciate the work the authors have done implementing additional methods (ML and Bayesian inference) and software (RAxML, MrBayes, IQtree). To include the IQtree and MrBayes trees in the supplement lends further credibility to the analysis.

The authors conclude that the low bootstrap values in their phylogenetic analyses are due to intrinsic sequence variation and not to the methods/programs used. I agree with the authors' conclusion, but regardless the reason, the authors need to be careful drawing strong conclusions based on clades with low bootstrap support. As the authors themselves point out, future sequencing of additional Streptophyte genomes will be required to resolve the relationships between the gene families. I add this only as a note; no further action needs to be taken.

L73 (former L75): the statement is still not correct. *Arabidopsis* DCL1, not AGO1, produces miRNAs. AGO1 is the major protein that interacts with miRNAs. I am sorry if my previous feedback was unclear; when I wrote "AGO1 does not produce miRNAs, it associates with them to get guided to a target RNA", I meant that miRNAs guide AGO1 to the target RNA.

L147 (former L148): you have not removed ref. 35.

L159: "AGO-like 18" could be misinterpreted as monocot-specific AGO18. But I assume that "18" refers to ref. 18. If so, please write "18" in superscript.

L204 (former L189): you have not removed ref. 35.

L240-243: *MIR* genes within introns is not necessarily the ancestral trait. This could also be due to convergent evolution in *Chlamydomonas* and animals. Besides, miRNAs might have evolved independently in the plant and animal lineages: see e.g., DOI: 10.1038/s41559-016-0027 and DOI: 10.1002/bies.201200055. Please also note that plant *MIR* genes are transcribed by Pol II, just like animal *MIR* genes.

L271 (former L245): you have not removed ref. 35.

L356: to me, it is odd to mention AGO2 in the context of AGO1/5/10 evolution. I suggest removing AGO2 from this line, especially since its antiviral role is correctly mentioned on line 359.

Table S2: thanks for updating Table S2! As a minor detail, *Selaginella moellendorffii* is "Selmo" in Table S1 and "Smog" in Table S2. Please use consistent abbreviations.

Cover art: AGO-like is still called "AGO4-like".

Reference list: please add journal titles to refs 49 and 62. There is also a spelling mistake in ref. 49. Correct is "Carrington JC".

Reviewer #3 (Remarks to the Author):

The authors addressed all comments on my part.

REVIEWERS' COMMENTS:

Reviewer #1 (Remarks to the Author):

The authors have improved the manuscript to a satisfactory level. Overall, I find that the authors have addressed the referees' comments in their manuscript and rebuttal letter. A few exceptions are listed below.

1. Thank you. I have no further comments.

Response: Thank you for your nice suggestions in the previous version of the manuscript, and now recommending the manuscript for publication.

2. I appreciate the work the authors have done implementing additional methods (ML and Bayesian inference) and software (RAxML, MrBayes, IQtree). To include the IQtree and MrBayes trees in the supplement lends further credibility to the analysis.

The authors conclude that the low bootstrap values in their phylogenetic analyses are due to intrinsic sequence variation and not to the methods/programs used. I agree with the authors' conclusion, but regardless the reason, the authors need to be careful drawing strong conclusions based on clades with low bootstrap support. As the authors themselves point out, future sequencing of additional Streptophyte genomes will be required to resolve the relationships between the gene families. I add this only as a note; no further action needs to be taken.

Response: Thanks for your kind understanding.

L73 (former L75): the statement is still not correct. *Arabidopsis* DCL1, not AGO1, produces miRNAs. AGO1 is the major protein that interacts with miRNAs. I am sorry if my previous feedback was unclear; when I wrote "AGO1 does not produce miRNAs, it associates with them to get guided to a target RNA", I meant that miRNAs guide AGO1 to the target RNA.

Response: We have now revised the sentence as "Arabidopsis has 10 AGO proteins, and previous studies have shown that miRNAs guide AGO1 to the target RNA"

L147 (former L148): you have not removed ref. 35.

Response: The reference 35 is removed now.

L159: "AGO-like 18" could be misinterpreted as monocot-specific AGO18. But I assume that "18" refers to ref. 18. If so, please write "18" in superscript.

Response: The mistake has been rectified now.

L204 (former L189): you have not removed ref. 35.

Response: The ref. 35 is removed now.

L240-243: *MIR* genes within introns is not necessarily the ancestral trait. This could also be due to convergent evolution in *Chlamydomonas* and animals. Besides, miRNAs might have evolved independently in the plant and animal lineages: see e.g., DOI: 10.1038/s41559-016-0027 and DOI: 10.1002/bies.201200055. Please also note that plant *MIR* genes are transcribed by Pol II, just like animal *MIR* genes.

Response: Yes, we do agree with this suggestion. But, our main emphasis was to show the difference about how the *MIR* genes are embedded “In plants, *MIR* genes are embedded within the noncoding sequences between protein-coding genes, while in *Chlamydomonas*, *MIR* genes are embedded within the introns of protein-coding genes”

L271 (former L245): you have not removed ref. 35.

Response: The ref. 35 is removed now.

L356: to me, it is odd to mention AGO2 in the context of AGO1/5/10 evolution. I suggest removing AGO2 from this line, especially since its antiviral role is correctly mentioned on line 359.

Response: We have removed AGO2 from this sentence.

Table S2: thanks for updating Table S2! As a minor detail, *Selaginella moellendorffii* is “Selmo” in Table S1 and “Smog” in Table S2. Please use consistent abbreviations.

Response: Thanks for pointing out the mistake. I have now rectified the mistake.

Cover art: AGO-like is still called “AGO4-like”.

Response: The mistake has been rectified now.

Reference list: please add journal titles to refs 49 and 62. There is also a spelling mistake in ref. 49. Correct is “Carrington JC”.

Response: The mistake has been rectified now.

Reviewer #3 (Remarks to the Author):

The authors addressed all comments on my part.

Response: Thank you for your nice suggestions in the previous version of the manuscript, and now recommending the manuscript for publication.